# Plutonium-Doped Monazite and Other Orthophosphates— Thermodynamics and Experimental Data on Long-Term Behavior

**Polina Mikhailova [1], Boris Burakov [2,3,*], Nikolai Eremin [1], Alexei Averin [4] and Andrey Shiryaev [4,*]**

1. Geology Department, Moscow State University, 119991 Moscow, Russia; mihaylowa.pol@yandex.ru (P.M.); neremin@mail.ru (N.E.)
2. V.G. Khlopin Radium Institute, 194021 St. Petersburg, Russia
3. A.F. Ioffe Institute, 194021 St. Petersburg, Russia
4. A.N. Frumkin Institute of Physical Chemistry and Electrochemistry RAS, 119071 Moscow, Russia; alx.av@yandex.ru
* Correspondence: burakov@peterlink.ru (B.B.); a_shiryaev@mail.ru (A.S.)

**Abstract:** The paper consists of two main parts: a microscopic and spectroscopic investigation of the single crystal of 17-year-old $^{238}$Pu-doped Eu-monazite, and a theoretical calculation of the properties of several structural types of orthophosphates. It is shown that actinide-doped monazite is prone to the formation of mechanically weak, poorly crystalline crust, presumably consisting of rhabdophane. Its formation is likely promoted by the formation of peroxides and, potentially, acidic compounds, due to the radiolysis of atmospheric moisture. The calculations of mixing the enthalpies and Gibbs energies of binary solid solutions of Pu and rare earth element (REE) phosphates that were performed for the principal structural types—monazite, xenotime, rhabdophane—show that, in the case of light REEs, the plutonium admixture is preferentially redistributed into the rhabdophane. This process strongly affects the behavior of actinides, leached from a monazite-based waste form. The applications of these results for the development of actinide waste forms are discussed. The current data on the behavior of real actinide-doped monazite suggest that this type of ceramic waste form is not very resistant, even in relatively short time periods.

**Keywords:** plutonium; monazite; rhabdophane; orthophosphate; radiolysis; nuclear waste

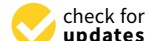



## 1. Introduction

The modeling of the long-term behavior of crystalline forms for actinides immobilization greatly benefits from investigation of the natural minerals comprising radioactive elements, such as uranium and thorium. Direct extrapolation of the results for the natural minerals to synthetic ceramics is complicated by markedly different concentrations of radionuclides and dose rate effects; nevertheless, some hints can be obtained. Monazite— ideally, a rare earth element phosphate (REEPO$_4$) with monoclinic structure—occupies a special place among actinide-containing minerals with remarkable chemical stability, as natural monazite is almost never present in metamict, that is, an amorphized state. Indeed, only very few works report diffuse X-ray diffraction patterns of natural Th-rich monazite [1]. Moreover, the samples studied in [1] were likely huttonite–monazite solid solutions, which are less radiation-resistant than pure monazite. The radiation damage in natural monazite is largely confined to small domains, eventually enriched in impurities [2,3]. A low critical temperature of recovery from radiation damage in monazite [4] leads to gradual annealing of natural grains and explains the lack of metamict samples (for review, see [5]), although the exact mechanism remains unclear [6]. Together with the remarkable flexibility of the monazite crystalline lattice and its chemical stability [7], these factors led to proposals of using monazite-based materials for the long-term immobilization of excess actinides [8–10].



However, to be industrially attractive, real ceramic waste forms should possess a reasonably high loading capacity to actinides, and realistic dose rates are several orders of magnitude higher than what is encountered in natural minerals containing admixtures of U and Th. Studies of radiation damage induced by ion beams or due to self-irradiation from isotopes with high specific activity, such as $^{238}$Pu or $^{244}$Cm, show that, in these conditions, monazite can readily be turned amorphous [4,5,11–13]. Moreover, although monazite is chemically rather stable, it is nevertheless subject to dissolution and other types of alterations. Cases of significant alteration of natural monazite with dramatic losses of Th, U, and radiogenic Pb are not uncommon (e.g., [14,15]).

An unusual type of alteration of $^{238}$Pu-doped single crystal Eu-monazite and polycrystalline La-monazite is observed on timescales of just several years of storage in ambient conditions: the formation of "blisters", or crust, which consists of Pu-containing rhabdophane (hydrated REE phosphate) with variable composition [16–18]. Its formation is almost certainly caused by intense damage and ionization by α-particles. The newly formed rhabdophane consists of submicron grains and is only poorly retained on the monazite surface. Whereas rhabdophane formation during a monazite-fluid interaction is a well-known phenomenon (e.g., [19]), its appearance, due to contact with atmospheric moisture, is unexpected, and certainly detrimental for eventual waste forms. Even although rhabdophane is poorly soluble [20], its nanocrystallinity may lead to the remobilization of actinides due to the destructive influence of high energy ions on nanoparticles [21]. Actinide-bearing nanoparticles are very mobile in geological environments and their migration is a subject of serious concern related to the safety of radioactive waste disposal. An investigation into the nanoparticles of LaPO$_4$ doped with α-emitting $^{225}$Ac showed that the retention of daughter products, including radioactive isotopes, by such nanoparticles is not very high; losses up to 50% were observed on a timescale of one month, which corresponds to $3T_{1/2}$ of the parent isotope (note the existence of several energetic decays in the $^{225}$Ac chain) [22]. The aim of the latter work differed from the examination of the structural changes of the nanoparticles and no relevant microscopic observations were performed. Some of the daughters may have been lost due to recoils; some might have been damaged, or they may have even exploded the nanoparticles following the mechanism described in [23]. In any case, the paper [22] demonstrates that even on moderate a timescale, monazite-based nanoparticles are not that resistant against loss of radionuclides.

This contribution concerns the behavior of Pu-containing REE orthophosphates and contains two independent but related parts. First, we present the experimental data behavior of aged (17-year-old) $^{238}$Pu-doped Eu-monazite single crystal. This part of the paper reveals the influence of the storage medium on the development of secondary mineralization on the surfaces of the actinide-doped monazite. In the second part, we describe the atomistic calculations of the thermodynamic functions, which reveal peculiarities of structural control on the behavior of the solid solution in Pu-REEPO$_4$ phases. The results of both experimental and modeling studies are discussed in the context of the application of monazite-based waste forms for the immobilization of actinides.

## 2. Materials and Methods

### 2.1. Samples and Experimental Details

Monazite single crystals with the composition of Eu$_{0.937}$Pu$_{0.063}$PO$_4$ doped with $^{238}$Pu (4.9 wt% $^{238}$Pu + 1.1 wt% of other Pu isotopes) were grown using the flux method in December 2003 at the V.G. Khlopin Radium institute [24]. All the samples were stored at 20–25 °C in ambient conditions. We note that, due to a rather short $T_{1/2}$ (87.7 y), the use of $^{238}$Pu allowed for the investigation of long-term effects from Pu decay in reasonably short periods of time, although the dose rate is much higher than in the case of, for example, $^{239}$Pu, and thus, direct comparison should be done with caution. The choice of europium as a matrix component was driven by the intention of synthesis of self-luminescing light sources and by the possibility of detecting minor changes in the matrix structure using Eu photoluminescence (PL) spectra. In addition, the crystallochemical behavior of Eu

is similar to that of Am, another actinide of high importance for the immobilization of fractionated waste.

In [16,17] we presented the results of the characterization of these samples of various ages (spanning in range from 11 to 15 years) using spectroscopy, X-ray, electron diffraction and absorption, and optical and electron microscopy. The samples preserved crystallinity, as shown by single crystal XRD and Raman scattering, but developed a whitish shell made of Pu-containing rhabdophane. There is strong evidence that the extent of the rhabdophane formation was influenced by the morphology of the monazite substrate [17].

In contrast to the samples studied in [16,17], the single crystal from the current work was hermetically sealed in a glass ampoule with ambient air several months after the synthesis. After 17 years of storage at 20–25 °C, the sample was studied using optical microscopy (Olympus BX51), and Raman and photoluminescence spectroscopy using a Renishaw inVia Reflex spectrometer. All studies were performed in situ in the ampoule; thus, interferences from the glass wall were present in Raman spectra. Lasers with 405 and 633 nm and long working distance objectives of $50\times$ and $100\times$ magnification were employed. Laser power was kept to a reasonable minimum and no changes in the sample under the probing beam were observed either visually or spectroscopically.

### 2.2. Atomistic Simulation

The semiempirical method of structural simulation described in detail in [25] was used for the theoretical estimation of the incorporation energies of plutonium impurities into the monazite (space group $P2_1/n$), xenotime ($I4_1/amd$), and anhydrous rhabdophane (C2) $TRPO_4$ structures (TR = Ln, Y). We note that, due to crystallochemical reasons, not all compounds mentioned above exist for every single rare earth element (REE), some of them are metastable or absent (e.g., [26]). Following [25], "virtual" phosphates were developed for these cases. The following effective charges were used: q (REE/Pu) = 1.6 $e_0$, q (P) = 1.2 $e_0$, and q (O) = $-0.7$ $e_0$. We employed Morse potentials developed in [25], which provided an excellent description of monazite and xenotime crystal structures and reproduced available experimental $S_0$(298 K) values with good accuracy:

$$(r) = D_M \cdot [\exp\left(-2\alpha\left(r - r_0\right)\right) - 2\exp\left(-\alpha\left(r - r_0\right)\right)] \tag{1}$$

The three variable parameters $D_M$ (eV), $\alpha$ ($\text{Å}^{-1}$), and $r_0$ (Å) denote, respectively, the energy of dissociation, the parameter of "softness" of the chemical bond, and the optimal interatomic distance in a contact pair. The parameters $D_M$, $\alpha$, $r_0$, and $R_{max}$ for light rare earth elements were optimized using the experimental structural data for the corresponding monazites from [27]. The parameters for heavy rare earth elements (Tb–Lu) were additionally optimized using the xenotime structural data from [28]. The Pu–O interaction parameters were optimized using the $PuPO_4$ monazite structural data from [27]. All the calculations were performed using GULP 4.1 and 4.5 software [29].

### 3. Results

### 3.1. Optical Microscopy and Spectroscopy

The gas inside the ampoule possessed a reddish-brown color. The Raman spectra of the gas phase contained two strong peaks centered at 750 and 1319 $\text{cm}^{-1}$, due to $NO_2$ [30]. The formation of nitrogen dioxide was also observed by us in a sealed ampoule with $^{238}Pu$-doped zircon of comparable age. The formation of $NO_2$ was due to the radiolysis of the sealed gases ([31] and refs. therein).

Optical photographs (Figure 1) of the studied monazite crystal were made in a hermetically sealed ampoule (see Section 2.1 for detail). The monazite crystal was rather uniformly covered by a thick white crust, which appeared to be considerably more developed in comparison with other samples from the same synthesis batch, which were stored in conditions of nonrestricted airflow [16,17]. The degree of the crust development and propensity to form bubble-like formations appeared to be determined by the morphology of the monazite crystal. The crust started cracking in some places, indicating brittle behavior. One apex

of the specimen broke away, revealing a dark brown color of the monazite core; however, light absorption in the crust may have altered the color perception.

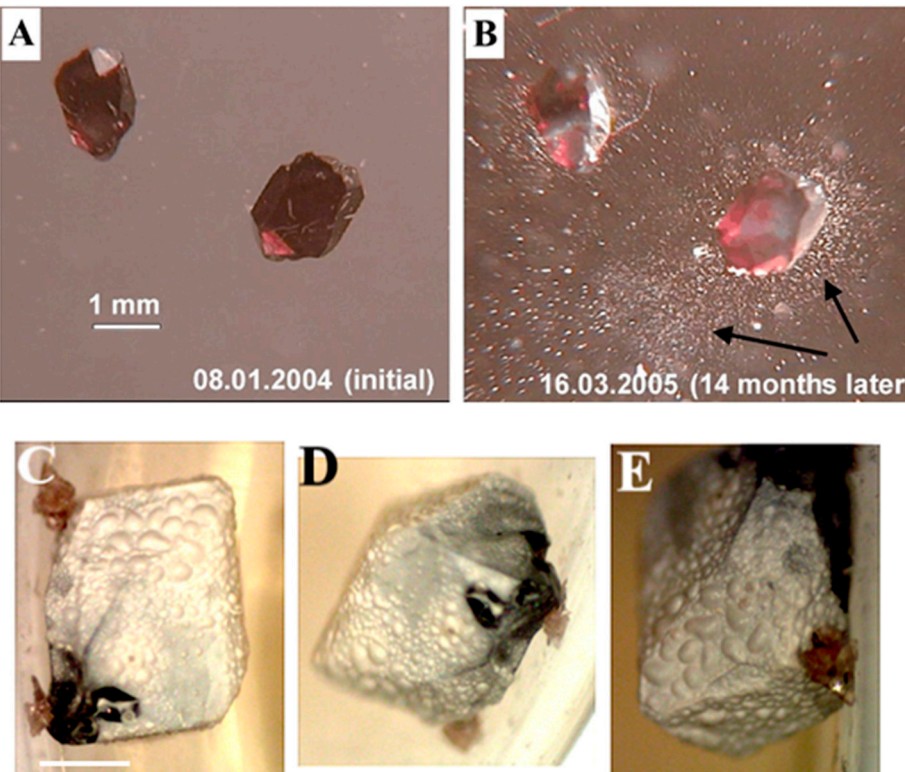

**Figure 1.** Optical photographs of the $^{238}$Pu-doped Eu-monazite crystal. (**A,B**): As-grown crystals and the same samples 14 months later [24], before sealing in the ampoule. Arrows in (**B**) show tiny particles spontaneously detached from the crystals. Scale bar is the same for (**A**) and (**B**), and is 1 mm. (**C–E**): Crystal in a sealed ampoule. Variations in the degree of development of the secondary phases on different crystallographic faces were observed. The brightness of the photographs (**C–E**) was increased to partly offset the brownish background due to $NO_2$. Scale bar is the same for (**C–E**) and is 0.5 mm.

The Raman and photoluminescence spectra of the shell and of the exposed core are shown in Figure 2, together with the representative spectra from our previous studies. The Raman spectra of the bulk monazite and crust (Figure 2b, Curves 1 and 2) were consistent with the monazite–rhabdophane system. However, in the crust, the intensity of the asymmetric P–O stretch in the phosphate groups (1045 cm$^{-1}$) was markedly enhanced and dominated the spectrum; the symmetric stretch was very weak. At present, we do not have an unambiguous explanation of this observation, but a strong distortion of the $PO_4^{3-}$ tetrahedra, and, possibly, protonation, are likely the causes.

In this work, the photoluminescence spectra of the crust were recorded using 405 nm excitation, which may have influenced the comparison with previous results. As expected, the PL spectra of the crust were dominated by the hypersensitive Eu $^5D_0 \rightarrow {}^7F_3$ transition, peaking at 617 nm. The transition was rather narrow and consisted of a single peak. The $^5D_0 \rightarrow {}^7F_4$ transition was very weak. The absence of the splitting of the $^5D_0 -> {}^7F_0$ peak (576 nm) indicates a single site for Eu$^{3+}$, but its exact symmetry remains uncertain due to the ambiguities in the analysis of the $^5D_0 -> {}^7F_1$ transition. In any case, the local environment of the europium ion in the crust formed on the specimen in a sealed ampoule differed from that in other studied samples.

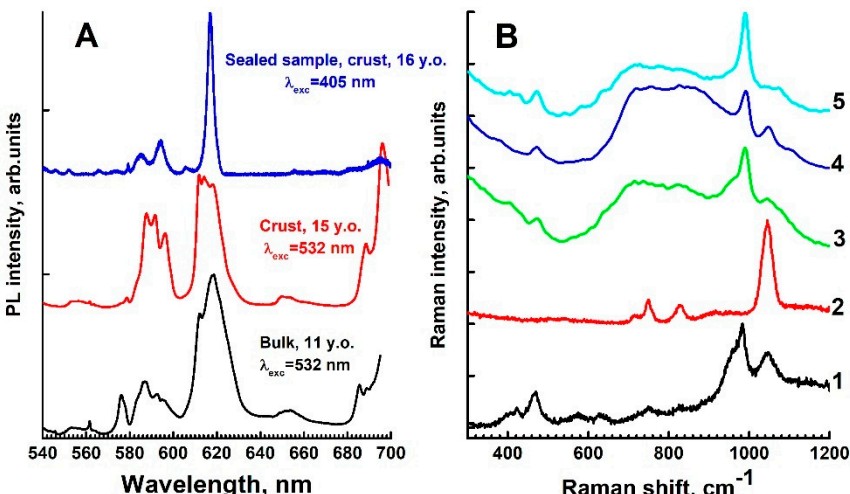

**Figure 2.** Spectra of the [238]Pu-doped Eu-monazite sample in comparison with some data from [16,17]. (**A**) photoluminescence spectra; excitation wavelength and age of the sample is indicated for every curve. (**B**) Raman spectra. Curve 1: core of the crystal from the present work (17 years old); 2: shell of the present crystal; 3: bulk of 15-year-old sample; 4: shell of 15-year-old crystal; 5: bulk of 11-year-old crystal. Curves 1, and 2 were recorded using 633 nm excitation; 3–5, with 532 nm excitation.

The principal results of the experimental study are confirmation of the rather rapid formation of secondary phases on the surfaces of the actinide-doped monazite samples observed in our previous works. This secondary phase is likely dominated by rhabdophane-like material, is mechanically unstable, as shown by the presence of cracks, and consists of submicron grains. An important difference of the current work is that it shows that the rhabdophane formation proceeds even in a closed system and is likely even accelerated in such an environment. The detrimental effect of the rhabdophane formation is emphasized by the preferential partitioning of Pu into this phase, as shown by the modeling results presented in the next section.

### 3.2. Thermodynamic Properties

3.2.1. Estimation of Interaction Energy and Enthalpy of Mixing

The reliability of the employed set of interatomic potentials and of the calculation approach as a whole is supported by a very good correspondence of the calculated and experimentally obtained crystal structures and the thermodynamic data of the monazites and xenotimes [25]. Using this set of potentials, the structural, elastic, and thermodynamic properties of the dehydrated rhabdophanes were calculated and summarized in Table 1.

**Table 1.** Structural, elastic, and thermodynamic characteristics of dehydrated rhabdophanes.

| Parameter | LaPO$_4$ | CePO$_4$ | PuPO$_4$ | PrPO$_4$ | NdPO$_4$ | SmPO$_4$ | SmPO$_4$ (Exp., [32]) | EuPO$_4$ | GdPO$_4$ |
|---|---|---|---|---|---|---|---|---|---|
| Volume, Å$^3$ | 594.05 | 580.70 | 573.78 | 571.41 | 562.57 | 545.70 | 540.97 (1) | 539.80 | 534.56 |
| a, Å | 12.6454 | 12.5428 | 12.4998 | 12.4710 | 12.4169 | 12.2862 | 12.1443 (1) | 12.2275 | 12.1754 |
| b, Å | 7.3008 | 7.2416 | 7.2167 | 7.2001 | 7.1689 | 7.0935 | 7.0178 (1) | 7.0596 | 7.0295 |
| c, Å | 6.4346 | 6.3933 | 6.3607 | 6.3637 | 6.3199 | 6.2614 | 6.3476 (1) | 6.2534 | 6.2459 |
| β,° | 90.00 | 90.00 | 90.00 | 90.00 | 90.00 | 90.00 | 90.02 (1) | 90.00 | 90.00 |
| Structural energy, eV | −34.314 | −34.582 | −34.687 | −34.802 | −35.220 | −35.506 | - | −35.568 | −35.648 |
| K, GPa | 94.58 | 93.01 | 39.26 | 39.86 | 46.17 | 52.69 | - | 50.98 | 50.01 |
| G, GPa | 41.94 | 43.29 | 45.38 | 44.05 | 47.50 | 50.82 | - | 49.83 | 49.18 |
| Q$_1$(TRPO$_4$-PuPO$_4$), kJ/mol | 0.964 | 0.097 | - | 0.039 | 0.264 | 1.767 | - | 2.687 | 3.624 |
| Q$_2$(PuPO$_4$-TRPO$_4$), kJ/mol | 0.987 | 0.101 | - | 0.039 | 0.263 | 1.745 | | 2.742 | 3.856 |
| ΔH$_{0,5}$ (TRPO$_4$-PuPO$_4$), kJ/mol | 0.2439 | 0.0248 | - | 0.0097 | 0.0658 | 0.4390 | - | 0.6786 | 0.9350 |

The mixing properties of the binary systems of rare-earth elements (TR = Ln, Y) with plutonium were calculated (Table 2). The Gibbs free energy of mixing was calculated using the following equation:

$$\Delta G_{mix} = \Delta H_{mix} - T\Delta S_{mix} = x_1 x_2 (x_2 Q_1 + x_1 Q_2) - T\Delta S_{mix} \qquad (2)$$

where $\Delta H_{mix}$ is the formation enthalpy at T = 0 K of a binary solid solution comprising molar fractions $x_i$ of the components. The interaction energies $Q_1$ and $Q_2$ can be calculated with the following equations:

$$Q_1 = E_{def} (TR2 \text{ in } TR1PO_4) + E_{str} (TR1PO_4) - E_{str} (TR2PO_4)$$

$$Q_2 = E_{def} (TR1 \text{ in } TR2PO_4) + E_{str} (TR2PO_4) - E_{str} (TR1PO_4)$$

where $E_{def}$ is the energy of an isolated defect consisting of $TR1$ isomorphous impurity in a phosphate of $TR2$; $E_{str}$ is the phosphate structural energy per formula unit.

Table 3 shows the enthalpy of mixing of the equimolar compositions of the solid solutions calculated using Equations (2) and (3) under the assumption that the interaction energy depends linearly on the composition of a solid solution. It is worth noting that the estimated values $\Delta H_{mix}$ strongly depend on $\Delta R$.

3.2.2. Calculation of the Properties of Mixing for Solid Solutions in Systems $LaPO_4$–$PuPO_4$, $EuPO_4$–$PuPO_4$, $GdPO_4$–$PuPO_4$ and $TbPO_4$–$PuPO_4$ by the Supercell Method

The Gibbs free energies in the systems $LaPO_4$–$PuPO_4$, $EuPO_4$–$PuPO_4$, $GdPO_4$–$PuPO_4$, and $TbPO_4$–$PuPO_4$ were calculated using the large supercell approach [33]. Three types of structure—monazite, xenotime, and rhabdophane—were considered. A $4 \times 4 \times 4$ supercell contains, depending on the structure, 192 to 256 isomorphic positions in the cationic sublattice (Figure 3). The choice of the elements is explained by the existence of our own experimental data for La and Pu [12,16–18]; Gd and Tb were selected as representatives of light (La–Gd) and heavy (Tb–Lu) lanthanides. The calculated Gibbs energies and mixing enthalpies are shown in Table 4.

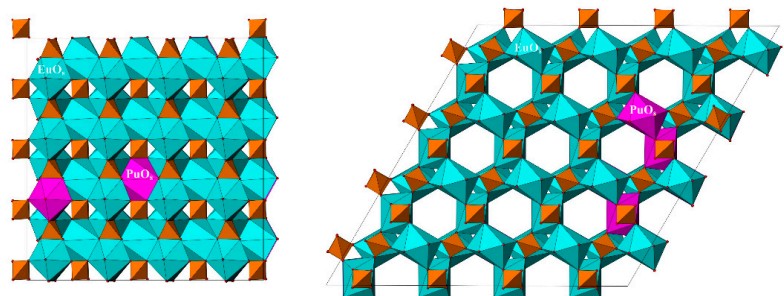

**Figure 3.** Rhabdophane $4 \times 4 \times 4$ supercell of $Eu_{182}Pu_{10}P_{192}O_{768}$ composition. Left: [100]; right: [001] plane.

The experimental [18,34] and theoretical [35,36] data suggest that the optimum amount of Pu in monazite-based waste forms should not exceed ~15 at.%. Based on this observation, the $TRPO_4$–$PuPO_4$ binary solid solutions containing from 5% to 15% plutonium were also studied. Supercells with four different atomic configurations representing a disordered solid solution as close as possible were used. The dependencies of the enthalpy of mixing on the composition of a solid solution for three polymorphic modifications of rare earth phosphates are shown in Figure 4. For all systems except $TbPO_4$–$PuPO_4$ with a xenotime structure, the values of mixing enthalpy obtained by the supercell method are systematically smaller for a small percentage than the values obtained in the approximation of infinite dilution [25]. Figure 5 shows the dependence of the obtained free energy values for solid solutions of Pu in REE phosphates with different structures and in a broad temperature range.

**Table 2.** Calculated energies of TR–Pu binary solid solutions of several considered structural types (monazite [25], xenotime, rhabdophane), calculated by the method of point defects in the limit of infinite dilution. The radii of the spheres used were 8.5 and 18.5 Å, respectively.

| Structure | Parameter | LaPO$_4$ | CePO$_4$ | PrPO$_4$ | NdPO$_4$ | SmPO$_4$ | EuPO$_4$ | GdPO$_4$ | TbPO$_4$ | DyPO$_4$ | YPO$_4$ | HoPO$_4$ | ErPO$_4$ | TmPO$_4$ | YbPO$_4$ | LuPO$_4$ |
|---|---|---|---|---|---|---|---|---|---|---|---|---|---|---|---|---|
| Monazite | $Q_1$ | 1.06 | 0.1 | 0.04 | 0.46 | 2.21 | 3.28 | 4.38 | 10.42 | 17.55 | 19.55 | 24.01 | 31.11 | 37.76 | 44.17 | 51.71 |
| | $Q_2$ | 1.06 | 0.12 | 0.05 | 0.46 | 2.15 | 3.13 | 4.16 | 10.08 | 16.00 | 17.71 | 21.36 | 27.18 | 32.58 | 37.71 | 43.66 |
| | η | 1.00 | 0.91 | 0.89 | 1.00 | 1.03 | 1.05 | 1.05 | 1.03 | 1.10 | 1.10 | 1.12 | 1.16 | 1.16 | 1.17 | 1.18 |
| Xenotime | $Q_1$ | 1.47 | 0.11 | 0.06 | 0.31 | 2.51 | 4.41 | 6.55 | 24.74 | 32.15 | 34.48 | 39.72 | 47.90 | 55.35 | 63.04 | 72.13 |
| | $Q_2$ | 1.44 | 0.11 | 0.06 | 0.31 | 2.46 | 4.19 | 6.04 | 18.79 | 23.98 | 25.76 | 29.19 | 34.78 | 39.81 | 44.90 | 50.82 |
| | η | 1.02 | 1.00 | 1.04 | 1.00 | 1.02 | 1.05 | 1.08 | 1.32 | 1.34 | 1.34 | 1.36 | 1.38 | 1.39 | 1.40 | 1.42 |
| Rhabdophane | $Q_1$ | 0.964 | 0.097 | 0.039 | 0.264 | 1.767 | 2.687 | 3.624 | 11.753 | 16.651 | 17.897 | 21.502 | 26.933 | 32.149 | 37.361 | 43.553 |
| | $Q_2$ | 0.987 | 0.101 | 0.039 | 0.263 | 1.745 | 2.742 | 3.856 | 12.655 | 16.288 | 17.609 | 20.299 | 24.833 | 29.092 | 33.343 | 38.344 |
| | η | 0.98 | 0.96 | 1.00 | 1.00 | 1.01 | 0.98 | 0.94 | 0.93 | 1.02 | 1.02 | 1.06 | 1.08 | 1.11 | 1.12 | 1.14 |

**Table 3.** The enthalpy of the mixing of equimolar compositions of binary solid solutions of rare earth and plutonium phosphates. Virtual minerals are marked with an asterisk.

| Monazite Solid Solution | $\Delta H_{0,5}$, kJ/mol | Xenotime Solid Solution | $\Delta H_{0,5}$, kJ/mol | Rhabdophane Solid Solution | $\Delta H_{0,5}$, kJ/mol |
|---|---|---|---|---|---|
| LaPO$_4$–PuPO$_4$ | 0.26 | LaPO$_4$ *–PuPO$_4$ * | 0.36 | LaPO$_4$–PuPO$_4$ * | 0.24 |
| CePO$_4$–PuPO$_4$ | 0.03 | CePO$_4$ *–PuPO$_4$ * | 0.03 | CePO$_4$–PuPO$_4$ * | 0.02 |
| PuPO$_4$–PrPO$_4$ | 0.01 | PuPO$_4$ *–PrPO$_4$ * | 0.01 | PuPO$_4$ *–PrPO$_4$ | 0.01 |
| PuPO$_4$–NdPO$_4$ | 0.12 | PuPO$_4$ *–NdPO$_4$ * | 0.08 | PuPO$_4$ *–NdPO$_4$ | 0.07 |
| PuPO$_4$–SmPO$_4$ | 0.54 | PuPO$_4$ *–SmPO$_4$ * | 0.62 | PuPO$_4$ *–SmPO$_4$ | 0.44 |
| PuPO$_4$–EuPO$_4$ | 0.80 | PuPO$_4$ *–EuPO$_4$ * | 1.08 | PuPO$_4$ *–EuPO$_4$ | 0.68 |
| PuPO$_4$–GdPO$_4$ | 1.07 | PuPO$_4$ *–GdPO$_4$ * | 1.57 | PuPO$_4$ *–GdPO$_4$ | 0.93 |
| PuPO$_4$–TbPO$_4$ * | 2.56 | PuPO$_4$ *–TbPO$_4$ | 5.44 | PuPO$_4$ *–TbPO$_4$ * | 3.05 |
| PuPO$_4$–DyPO$_4$ * | 4.19 | PuPO$_4$ *–DyPO$_4$ | 7.02 | PuPO$_4$ *–DyPO$_4$ * | 4.12 |
| PuPO$_4$–YPO$_4$ * | 4.66 | PuPO$_4$ *–YPO$_4$ | 7.53 | PuPO$_4$ *–YPO$_4$ * | 4.44 |
| PuPO$_4$–HoPO$_4$ * | 5.67 | PuPO$_4$ *–HoPO$_4$ | 8.61 | PuPO$_4$ *–HoPO$_4$ * | 5.23 |
| PuPO$_4$–ErPO$_4$ * | 7.29 | PuPO$_4$ *–ErPO$_4$ | 10.34 | PuPO$_4$ *–ErPO$_4$ * | 6.47 |
| PuPO$_4$–TmPO$_4$ * | 8.79 | PuPO$_4$ *–TmPO$_4$ | 11.90 | PuPO$_4$ *–TmPO$_4$ * | 7.66 |
| PuPO$_4$–YbPO$_4$ * | 10.23 | PuPO$_4$ *–YbPO$_4$ | 13.49 | PuPO$_4$ *–YbPO$_4$ * | 8.84 |
| PuPO$_4$–LuPO$_4$ * | 11.92 | PuPO$_4$ *–LuPO$_4$ | 15.37 | PuPO$_4$ *–LuPO$_4$ * | 10.24 |

**Table 4.** Mixing enthalpies and Gibbs free energies in systems $LaPO_4$–$PuPO_4$, $EuPO_4$–$PuPO_4$, $GdPO_4$–$PuPO_4$, and $TbPO_4$–$PuPO_4$ at a Pu fraction of 10%.

| | $\Delta H_{0,1}$ kJ/mol | $\Delta G_{0,1}$ (200 K) kJ/mol | $\Delta G_{0,1}$ (400 K) kJ/mol | $\Delta G_{0,1}$ (600 K) kJ/mol |
|---|---|---|---|---|
| | $LaPO_4$–$PuPO_4$ system | | | |
| Monazite | 0.085 | −0.4588 | −1.0027 | −1.5473 |
| Xenotime | 0.119 | −0.4271 | −0.9743 | −1.5397 |
| Rhabdophane | 0.067 | −0.4722 | −1.0113 | −1.5505 |
| | $EuPO_4$–$PuPO_4$ system | | | |
| Monazite | 0.268 | −0.2579 | −0.7845 | −1.3117 |
| Xenotime | 0.367 | −0.1820 | −0.7335 | −1.2850 |
| Rhabdophane | 0.194 | −0.3457 | −0.8862 | −1.4269 |
| | $GdPO_4$–$PuPO_4$ system | | | |
| Monazite | 0.376 | −0.1529 | −0.6819 | −1.2111 |
| Xenotime | 0.533 | −0.0235 | −0.5833 | −1.5726 |
| Rhabdophane | 0.269 | −0.2764 | −0.8224 | −1.3684 |
| | $TbPO_4$–$PuPO_4$ system | | | |
| Monazite | 0.867 | 0.3088 | −0.2351 | −0.7787 |
| Xenotime | 2.068 | 1.4401 | 0.8101 | −1.7786 |
| Rhabdophane | 0.897 | 0.8257 | 0.2557 | −0.3034 |

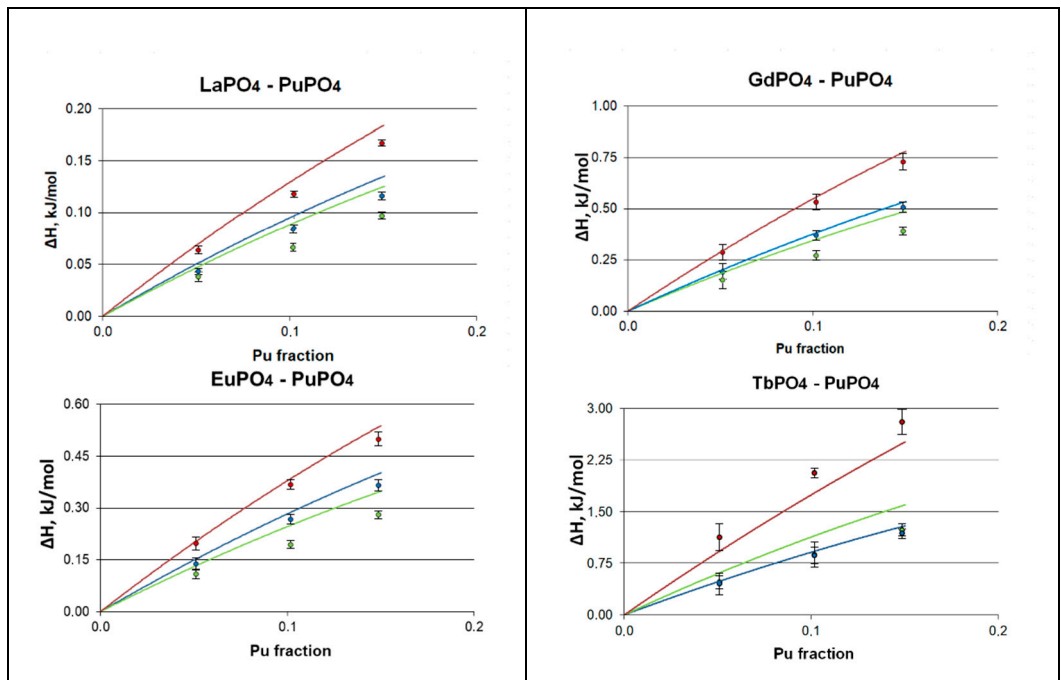

**Figure 4.** Enthalpy of mixing of the systems $LaPO_4$–$PuPO_4$, $EuPO_4$–$PuPO_4$, $LaPO_4$–$PuPO_4$, $GdPO_4$–$PuPO_4$, and $TbPO_4$–$PuPO_4$ as a function of Pu content. Symbols: calculation with the $4 \times 4 \times 4$ supercell model for different atomic configurations; mean square deviations are shown. Solid lines: calculation of the infinite dilution approximation.

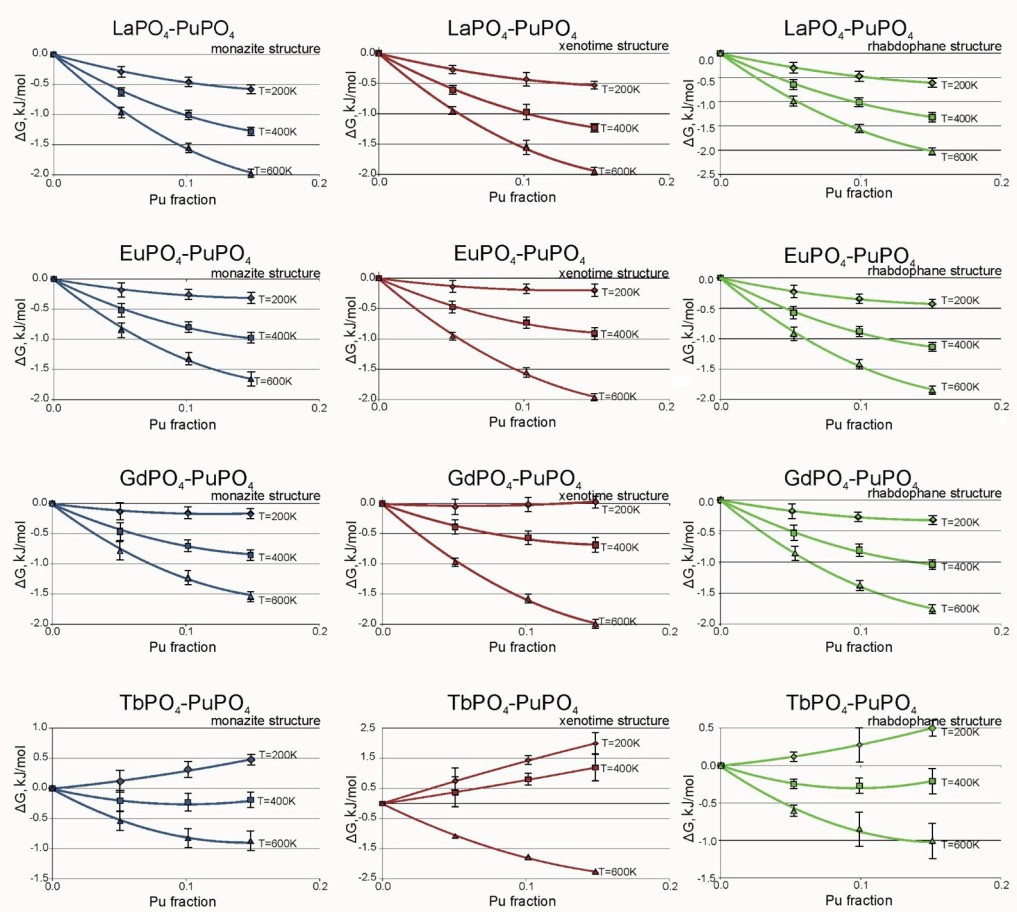

**Figure 5.** Free energies of solid solution for LaPO4–PuPO4, EuPO4–PuPO4, GdPO4–PuPO4, and TbPO4–PuPO4 systems as a function of Pu content. The calculations were performed for monazite, xenotime, and rhabdophane structures at 200, 400, and 600 K. The mean square deviations of values for different atomic configurations are shown. The lines are drawn to guide the eye.

## 4. Discussion

The results of the present work provide important information about the potential use of monazite-based ceramics or compounds for the long-term immobilization of actinides. Whereas the investigation of natural samples and extensive laboratory studies of the broad range of non-radioactive and low-radioactive synthetic REE-monazite ceramics and crystals gave promising results, the experiments with industry-relevant levels of doping with actinides show that chemical and radiation stability of highly radioactive material is not sufficiently high.

Comparison of the present work with our previous studies suggests that rhabdophane formation on the surfaces of actinide-doped monazite crystals is stimulated not only (or solely) by the interaction with atmospheric moisture, but is markedly enhanced by radiolysis effects. It is reasonable to suggest that the intense ionization in the vicinity of a Pu-doped monazite surface, either external or freshly exposed by mechanical cracking due to swelling, may produce, not only $NO_2$ observed by us in significant amounts, but other oxidizers, such as hydrogen peroxide or even some acidic compounds as well. We note here that the treatment of monazite with hot concentrated acids is a common approach to release REEs, Th, and U (monazite "cracking", e.g., [37]). The investigation into the synthetic La-monazite leaching in acidic media [38] indicated the formation of a thin rhabdophane layer, which controls subsequent processes.

The marked dependency of the extent of the secondary phase formation on crystallographic orientation is not yet fully understood. It may be caused by preferential

incorporation of a Pu impurity in different growth sectors (see discussion in [17]), or reflect a kind of epitaxial relationship. This issue presents certain applied interest, since the mechanical properties of La-monazite ceramics notably depend on the morphology of comprising grains [39]. Moreover, some important thermodynamic properties of monazite (e.g., thermal capacity) also demonstrated marked morphological effects [40]. Consequently, an understanding of the crystallographic control of the monazite alteration may help to reduce its impact on the waste form stability.

According to the results presented in Section 3.2, the formation of the rhabdophane phase on the monazite-based actinide-loaded waste form is not that benign. The enthalpy of mixing and the Gibbs free energy of Pu-REEPO$_4$ solid solutions with various structures indicate that, for compounds of light REE, the plutonium tends to redistribute into the rhabdophane phase; for heavy REE, orthophosphates monazite is the main Pu host. Rhabdophane is generally less stable against chemical alteration, and thus, its propensity to concentrate actinides is clearly detrimental. Even recalling that rhabdophane may transform back into monazite upon heating, the thermodynamically-favored concentration of Pu in the former phase may lead to the formation of free Pu-phase, for example, an oxide, readily forming mobile colloids. Together with the obvious nanocrystallinity of the rhabdophane, the eventual Pu-phase may easily lead to the remobilization of an actinide from the waste form and their near- and far-field migration.

## 5. Conclusions

We present new experimental and computational results relevant for the understanding of the long-term behavior of monazite-based waste forms loaded with several atomic percentages of plutonium. It has been shown that conclusions about the high performance of monazite waste forms based on the studies of simulant materials, that is, those with rare earth elements only, are too optimistic. It is usually assumed that a waste form will experience degradation only in the relatively distant future, when engineering barriers and containers will (partly) fail and fluids will be in contact with the form. We have shown that the marked degradation of Pu-doped monazite ceramics and even single crystals readily occurs even during storage in air. Presumably, the radiolysis of atmospheric moisture and the surrounding air by $\alpha$-particles dramatically accelerates monazite alteration. This process might be slowed down by the storage of the waste forms in inert, dry atmospheres, but such an approach will make the repository even more costly and will set new criteria for hermeticity of the canisters on very long timescales. Thermodynamic calculations suggest that, in the case of orthophosphates of light, REE plutonium favors less stable rhabdophane structures. Recalling that rhabdophane almost always consists of mechanically fragile and, importantly, micron- and submicron crystallites, the rapid formation of rhabdophane and/or other secondary phases demonstrated in our studies raises major concerns about the eventual large-scale use of monazite-based forms in real life.

**Author Contributions:** P.M., N.E.—thermodynamic calculations and writing; B.B.—synthesis and storage of samples; A.A.—spectroscopic measurements; A.S.—conceptualization, writing. All authors have read and agreed to the published version of the manuscript.

**Funding:** Experimental part of this research was funded by the Ministry of Science and Higher Education (grant agreement no. 075–15–2020–782).

**Institutional Review Board Statement:** Not applicable.

**Informed Consent Statement:** Not applicable.

**Data Availability Statement:** Not applicable.

**Conflicts of Interest:** The authors declare no conflict of interest.

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
