# Peer review of "Plutonium-Doped Monazite and Other Orthophosphates—Thermodynamics and Experimental Data on Long-Term Behavior"

_sustainability, doi:10.3390/su13031203_

Round 1

Reviewer 1 Report

The authors submitted the manuscript entitled Plutonium-Doped Monazite and Other Orthophosphates – Thermodynamics and  Experimental Data on Long-Term Behavior.

The behavior of highly radioactive materials is one of the long-standing issues in science and technology, particularly, when considering the durability and long-term stability of materials.

The authors combined experimental and theoretical approaches.

In general, the study is valuable and seems to be interesting for readers.

Here is a list of minor problems:

The details of equipment should be added in the Materials and Methods section.

In Fig.1 scale bar should be added in every picture (A-E).

The authors should care about the upper/lower case and italic/non-italic for parameter symbols and units. For example: in Figures 4 and 5, KJ/mol should be kJ/mol; Q in equation 2 and lines 161,162, etc.

The units of parameters should be provided in tables.  

Author Response

The Reviewer has raised the following questions.

Q1 )The details of equipment should be added in the Materials and Methods section.

REPLY. The corresponding information is added.

Q2) In Fig.1 scale bar should be added in every picture (A-E).

REPLY. The scale bar was added. Note that the size of the bar is equal for panels A-B and C-E and thus corresponding note is added to the figure caption. Adding a separate bar to, say, panel B will interfere with some details of the image.

Q3) The authors should care about the upper/lower case and italic/non-italic for parameter symbols and units. For example: in Figures 4 and 5, KJ/mol should be kJ/mol; Q in equation 2 and lines 161,162, etc. The units of parameters should be provided in tables.

REPLY. These issues are corrected.

Reviewer 2 Report

The authors need to consider the experimental error (Figs. 4 and 5.)

I do not see any comparison between the experimental results and the modelling.

The authors need to include paragraphs about the deeper application of the system.

Author Response

The following questions were raised.

Q1) The authors need to consider the experimental error (Figs. 4 and 5.)

REPLY. The errors are added.

Q2) I do not see any comparison between the experimental results and the modelling.

REPLY. This is a surprising and unclear comment. In our manuscript we first show formation of a rhabdophane shell on surface of monazite crystal and discuss differences of behaviour of samples stored in air and in sealed ampoule. We explicitly discuss why rhabdophane formation is detrimental for waste forms.

The modeling part shows that Pu will partition into the rhabdophane phase. Consequently, both parts, experimental and theoretical one, address the same issue of (questionable) applicability of monazite as a reliable waste form.

Q3) The authors need to include paragraphs about the deeper application of the system.

REPLY. This is even more confusing question. The introduction, discussion and conclusion sections include justification of the current work and provide necessary references. Both reviews and original works are cited. We may, of course, add some hands-waiving, but the manuscript is NOT about philosophy. The intended audience is already familiar with the immobilisation problem.

Round 2

Reviewer 2 Report

The comparison between the experiment and theory:

  • I do not see any direct comparison between the experiment and the theory
  • If possible it would be very good to have a graph or table with both experimental and theoretical data. Please mark this comparison red in the text that one can see

If the comparison is already made just color it that I see.

Author Response

As we have emphasized in previous version that comparison of experimental and modelling data in NOT the aim of this paper. One of the reasons is that despite soundness of the experimental observations made by us before and enhanced by results described here, too many important, but insufficiently understood details preclude reliable theoretical description.

Reading of the manuscript makes it obvious that is consists of two independent, but related parts.

In the first, experimental part, we report new data on behaviour of Pu-doped monazite. A significant difference of the presented data relative to previous works is examination of influence of the environment on formation of secondary phase. So, we SHOW formation of rhabdophane.

The second, modeling part, explains WHY rhabdophane is detrimental. Again, if one really looks at the results, he will even see that compositions close to experimental ones, are described in the modeling part.

In the revised version of the manuscript we have added the following paragraphs:

"The paper consists of two main parts: microscopic and spectroscopic investigation of single crystal of 17 years old 238Pu-doped Eu-monazite, and theoretical calculation of properties of several structural types of orthophosphates. It is shown that actinide-doped monazite are prone to formation of mechanically weak, poorly crystalline crust, presumably consisting of rhabdophane. Its formation is likely promoted by formation of peroxides and, potentially, acidic compounds due to radiolysis of atmospheric moisture. Calculations of mixing enthalpies and Gibbs energies of binary solid solutions of Pu and REE phosphates performed for principal structural types – monazite, xenotime, rhabdophane show that in case of light REE the plutonium admixture is preferentially redistributed into the rhabdophane. This process will strongly affect behavior of actinides, leached from a monazite-based waste form."

"This contribution concerns behavior of Pu-containing REE orthophosphates and contains two independent, but related parts. First, we present experimental data behavior of aged (17 years old) 238Pu-doped Eu-monazite single crystal. This part of the paper reveals influence of storage medium on development of secondary mineralization on surfaces of the actinide-doped monazite. In the second part, we describe atomistic calculations of thermodynamic functions, which reveal peculiarities of structural control on behavior of solid solution in Pu-REEPO4 phases. Results of both experimental and modeling studies are discussed in context of application of monazite-based waste forms for immobilization of actinides."

 "The principal results of the experimental study is confirmation of rather rapid formation of secondary phases on surfaces of actinide-doped monazite samples observed in our previous works. This secondary phase is likely dominated by rhabdophane-like material, is mechanically unstable as shown by presence of cracks, and consists of submicron grains. An important difference of the current work is that it shows that the rhabdophane formation proceeds even in closed system and is likely even accelerated in such environment. Detrimental effect of the rhabdophane formation is emphasized by preferential partitioning of Pu into this phase, as shown by the modeling results presented in the next section."

Round 3

Reviewer 2 Report

The changes were introduced. The manuscript can be accepted.